# Therapeutic Vaccinations with p210 Peptides in Imatinib-Treated Chronic Myeloid Leukemia Patients: 10 Years Follow-Up of GIMEMA CML0206 and SI0207 Studies

**DOI:** 10.3390/vaccines13040419

**Published:** 2025-04-16

**Authors:** Anna Sicuranza, Massimo Breccia, Francesco Iuliano, Gabriele Gugliotta, Fausto Castagnetti, Monia Lunghi, Andrea Patriarca, Tamara Intermesoli, Luigiana Luciano, Antonella Russo Rossi, Giovanna Rege Cambrin, Vladan Vucinic, Michele Malagola, Alessandra Malato, Elisabetta Abruzzese, Mariella D’Adda, Sara Galimberti, Marzia Defina, Vincenzo Sammartano, Cristiana Cafarelli, Emanuele Cencini, Alessandra Cartocci, Paola Pacelli, Alfonso Piciocchi, Arianna Rughini, Dietger Niederwieser, Monica Bocchia

**Affiliations:** 1Hematology Unit, Azienda Ospedaliera Universitaria Senese, University of Siena, 53100 Siena, Italy; marziadefina@libero.it (M.D.); vincenzo.sammartano2@gmail.com (V.S.); cristianacafarelli21@gmail.com (C.C.); cencioema@libero.it (E.C.); paola.pacelli@unisi.it (P.P.); monica.bocchia@unisi.it (M.B.); 2Department of Translational and Precision Medicine, Sapienza University, 00161 Rome, Italy; breccia@bce.uniroma1.it; 3Presidio Ospedaliero N. Giannetasio, Azienda ASL 3, 87068 Rossano, Italy; iuliano54@outlook.it; 4Institute of Hematology “L. and A. Seràgnoli”, Department of Experimental, Diagnostic and Specialty Medicine, “S. Orsola-Malpighi” University Hospital, University of Bologna, 40138 Bologna, Italy; gabriele.gugliotta@unibo.it (G.G.); fausto.castagnetti@unibo.it (F.C.); 5Division of Hematology, Department of Translational Medicine, University of Eastern Piedmont and AOU Maggiore della Carità, 28100 Novara, Italy; monia.lunghi@med.uniupo.it (M.L.); andrea.patriarca@uniupo.it (A.P.); 6Hematology and Bone Marrow Transplant Unit, ASST Papa Giovanni XXIII, 24127 Bergamo, Italy; tintermesoli@asst-pg23.it; 7Hematology Unit “Federico II”, University of Naples, 80138 Naples, Italy; lulucian@unina.it; 8Hematology and Stem Cell Transplantation Unit, Department of Precision and Regenerative Medicine and Ionian Area (DiMePRe-J), University of Bari “Aldo Moro”, 70121 Bari, Italy; antonella.russorossi@gmail.com; 9Division of Internal Medicine and Hematology, San Luigi Gonzaga Hospital, 10043 Turin, Italy; giovanna.rege@libero.it; 10Klinik und Poliklinik für Hämatologie, Zelltherapie und Hämostaseologie, University Hospital Leipzig, 04103 Leipzig, Germany; vladan.vucinic@medizin.uni-leipzig.de (V.V.); dietger.niederwieser@medizin.uni-leipzig.de (D.N.); 11Department of Clinical and Experimental Sciences, University of Brescia, Unit of Blood Diseases and Bone Marrow Transplant, ASST Spedali Civili, 25123 Brescia, Italy; michele.malagola@unibs.it; 12A.O. Ospedali Riuniti Villa Sofia-Cervello-P.O. Cervello, 90146 Palermo, Italy; alessandramalato@hotmail.com; 13Division of Hematology, S. Eugenio Hospital, 00144 Rome, Italy; elisabetta.abruzzese@uniroma2.it; 14Division of Hematology, ASST-Spedali Civili di Brescia, 25123 Brescia, Italy; marielladadda@libero.it; 15Section of Hematology, Department of Clinical and Experimental Medicine, University of Pisa, 56126 Pisa, Italy; sara.galimberti@unipi.it; 16Department of Medical Sciences, Surgery and Neurosciences, University of Siena, 53100 Siena, Italy; alessandra.cartocci@dbm.unisi.it; 17GIMEMA Foundation, 00182 Rome, Italy; a.piciocchi@gimema.it (A.P.); a.rughini@gimema.it (A.R.)

**Keywords:** CML, peptide, vaccine, imatinib, immune therapy

## Abstract

**Background:** We previously showed that peptides encompassing the unique b3a2 or b2a2 breakpoint amino-acid sequence of oncogenic p210 induced peptide-specific T-cell responses in chronic myeloid leukemia (CML) patients. **Methods**: From 2007 to 2011, two multicenter peptide vaccine phase II studies, GIMEMA CML0206 and SI0207, enrolling overall 109 CML patients (68 b3a2 and 41 b2a2) with persistence of molecular disease during imatinib treatment, were carried out. Peptide vaccination schedule included the following: “immunization phase” (six vaccinations every 2 weeks); “reinforcement” phase (three monthly boosts) and “maintenance” phase (two boosts at 3-month intervals). GM-CSF (granulocyte-macrophage-colony-stimulating factor, sarmograstim) served as the immunological adjuvant. **Results**: The short-term results (at completion of vaccine protocol—12 months) and long-term follow-up are reported. All patients completed the vaccination schedule with no toxicity. After vaccinations, the BCR::ABL1 peptide-specific CD4+ T-cell response was documented in 80% of patients. In the short term, 30% of patients achieved a reduction in BCR::ABL1, while the majority showed stable molecular disease with fluctuations. The median follow-up since diagnosis and last vaccination are 18 and 10 years, respectively, with an overall survival (OS) rate at 18 years of 89%. In addition, 97/109 (89%) patients are alive, while 12/109 (11%) died of CML-unrelated reasons. Overall, 18/109 (16.5%) patients are in treatment-free remission (TFR) for a median time of 48 months. **Conclusions**: The long-term results of p210 peptide vaccinations in CML patients with persisting disease during imatinib treatment showed its feasibility, safety, absence of off-targets events, high OS and not negligible rate of successful TFR. Active immunotherapeutic approaches in CML patients with low disease burden, eventually employing newer vaccine strategies such as mRNA vaccines, may be reconsidered.

## 1. Introduction

Chronic myeloid leukemia (CML) has pioneered the era of precision medicine, being the first human cancer for which a “magic bullet” specifically targeting p210, the oncogenic protein derived by chimeric BCR::ABL1 fusion gene with constitutive tyrosine kinase activity causing the disease, was developed [1]. In 2001, the introduction of tyrosine kinase inhibitors (TKIs), led by imatinib mesylate, modified the management of CML patients and dramatically improved their life expectancy [1,2,3], with a median overall survival, during chronic TKI treatment, comparable to that of the general population. Despite these exceptional results, today, only a minority of CML patients experience treatment-free remission (TFR), the best surrogate of “cure” in CML, while most of them undergo a TKI life-long treatment [4]. Indeed, quiescent CML leukemia stem cells (LSCs) survive despite BCR::ABL1 inhibition by TKIs [5], representing a disease “reservoir” still detectable in patients in complete cytogenetic response (CCyR) [6], deep molecular response (DMR) and, surprisingly, even in stable TFR [7]. The latter is in favor of an immunological control of residual CML LSCs that prevents overt disease recurrence, although detailed mechanisms of how the immune system exerts its control are yet unclear [8].

In the pre-imatinib era, an alternative immune-based target therapy was hypothesized based on the potential immunogenicity of p210 in virtue of the unique sequence of amino acids contained in its junctional region representing a tumor-specific determinant [9]. Indeed, after screening the ability of a series of synthetic peptides corresponding to the junction sequences of p210 protein to bind according to putative anchor motifs class I and class II molecules (i.e., HLA A1, A2.1, A3.2, B8 and DR11), we and others identified some b3a2 and b2a2 breakpoint peptides able to elicit in vitro a specific T cell response both in normal donors and in CML patients [9,10]. Additionally, the evidence that CML leukemic cells present HLA-associated immunogenic peptides derived from the BCR::ABL1 was later confirmed by Clark et al. [11].

As such, in 2005, given the fact that, albeit its impressive results, imatinib was not able to eradicate CML, we developed a peptide vaccine (CMLVAX100) containing five p210/b3a2 breakpoint-derived peptides (four short peptides binding to HLA class I molecules and one long peptide binding to HLA class II molecules) associated with QS-21 (derived from the tree Quillaja Saponaria) and low doses of GM-CSF as immunological adjuvants. The intent of peptide vaccinations was to further reduce minimal residual disease (MRD) in imatinib-treated CML patients [12].

In this pivotal phase II study in ten chronic-phase CML (CP-CML) patients presenting a stable cytogenetic and/or molecular residual disease during imatinib, we observed a substantial disease reduction after vaccinations, with some patients even gaining a complete molecular response [12]. Clinical improvement was associated with a vaccine-induced immune response, mainly consisting of peptide-specific CD4+ T cell proliferation. None of the patients experienced any toxicity other than local mild pain, redness and itching at the site of vaccination. No systemic adverse events and no severe adverse events were recorded [12]. Based on this pivotal experience, we conducted two consecutive prospective multicenter phase II studies, GIMEMA CML0206 (including five b3a2-derived peptides) and SI0207 (including one b2a2-derived peptide) in order to explore the impact of a peptide vaccine-based active specific immunotherapy on residual disease surviving imatinib. Here, we present the outcome of 109 CP-CML patients vaccinated with p210 specific peptide-vaccine while on imatinib treatment after a median observation time of 10 years.

## 2. Materials and Methods

### 2.1. Patient Eligibility

CP-CML patients treated with imatinib for at least 18 months, in stable CCyR but with persistence of molecular residual disease, entered either of the two phase II prospective, not-randomized multicenter trials. Patients with b3a2 breakpoint were enrolled in the GIMEMA CML0206 study, while patients with b2a2 entered the SI0207 study. No HLA selection was applied at enrolment. The exclusion criteria included concomitant use of immunosuppressive agents and the presence of immune deficiency or autoimmune disorders. Both studies were approved by the local Ethical Committee and Istituto Superiore di Sanità (GIMEMA0206 Protocol ID EUDRACT-2006-006189-40; SI0207 Protocol ID EudraCT 2008-001107-27).

### 2.2. Study Design and Treatment

Clinical and molecular evaluation of enrolled patients was performed according to standard guidelines for imatinib-treated CML patients [2]. In the GIMEMA CML0206 trial, b3a2 CP-CML patients fulfilling the inclusion criteria received the CMLVAX100 vaccine (5 peptides); in the SI0207 trial eligible, b2a2 CP-CML patients received the CMLVAXb2a2-25 vaccine (1 peptide). Additional details on peptide sequences and manufacturing of vaccines are described in the Appendix A. In both vaccine studies, GM-CSF (granulocyte-macrophage-colony-stimulating factor, sarmograstim) served as the immunological adjuvant. The peptide vaccination schedule consisted of 6 vaccinations at 2-week intervals (“immunization” phase), followed by 3 monthly boosts of vaccine (“reinforcement” phase) and subsequently by 2 additional boosts at a 3-month interval (“maintenance” phase) (Figure 1). The day before and the day of peptide vaccination, the patients subcutaneously received GM-CSF at a 50 µg/dose close to the vaccination site. After each vaccination, the patients were monitored for any local or general toxicity, as well for compliance with the treatment. All patients continued imatinib treatment during the vaccination protocol.

### 2.3. Endpoints

The primary endpoint of both the GIMEMA CML0206 and the SI0207 trial was to assess the rate of patients showing a measurable decrease in the peripheral blood BCR::ABL1/ABL1 ratio (expressed both as a copy number and log reduction) compared to individual pre-vaccine levels. The secondary endpoints were the rate of vaccine discontinuation because of toxicity and the rate of positive in vitro peptide-specific immune response induced by the vaccinations.

### 2.4. Peptide Vaccine Induced Immune Response Evaluation “In Vitro”

Based on our pilot vaccination study [12], we decided to measure the immunological effect of peptide vaccination by using a peptide-specific CD4+ T cell proliferation assay against the long HLA-class II binder peptide included in each vaccine.

As such, IVHSATGFKQSSKALQRPVASDFEP (b3a2-25 peptide) and TVHSIPLTINKEEALQRPVASDFEP (b2a2-25 peptide) peptide-specific CD4+ T cell proliferation was measured at baseline and at various time points after vaccinations to assess the immune response to CMLVAX100 and CMLVAXb2a2, respectively.

Briefly, at each timepoint, freshly drawn, purified (by the magnetic bead separation Miltenyi Biotec standard assay) CD4+ cells were incubated for 96 h at 37 °C with 20 µg per mL of peptides under the following conditions: (1) in the presence of the IVHSATGFKQSSKALQRPVASDFEP peptide (for patients enrolled in the CML0206 study and vaccinated with CMLVAX100); (2) in the presence of the peptide TVHSIPLTINKEEALQRPVASDFEP (for patients enrolled in the SI0207 study and vaccinated with CMLVAXb2a2-25); (3) in the presence of five control peptides (including HLA-A3 and HLA-A11 binding sequences from HIV, a HLA-A2 binding sequences from influenza, PR3-derived 25-amino acids peptide class II binder); and (4) in the absence of peptides. CD4+ T cell peptide-specific proliferation was measured by using a standard 3H thymidine incorporation assay. The results are expressed in counts per minute (cmp) and as the mean of the quintuplicate wells for each condition.

The proliferation results are expressed in stimulation index values (S.I.), calculated as follows: (S.I. = counts per minute in the test sample (i.e., CD4+ T cells plus peptides)/counts per min in the control sample (i.e., CD4+ T cells alone). Based on our previous preliminary data [12], in which, after six vaccinations, the CML patients showed a varying specific proliferation response in the presence of all five CMLVAX100 peptides with a stimulation index range between 1.8 and 25, a positive peptide-specific CD4+ T cell proliferation was considered as a value of S.I. ≥ 1.8.

### 2.5. Molecular Response Evaluation

Molecular monitoring of BCR::ABL1 transcript levels in peripheral blood during the vaccine study core was carried out through quantitative RT-PCR assessments, according to the guidelines employed at that time [13,14] expressed both as a copy number of BCR::ABL1 transcript and log reduction from individual baseline values. The major molecular response was defined as a 3-log reduction from baseline (MR3 or MMR, corresponding to BCR::ABL1 IS ≤ 0.1%). At study entry, the patients were categorized as non-MR3 (less than 3-log reduction from individual baseline) MR3, and >MR3 (more than 3-log reduction). Later, follow-up molecular evaluation was defined according to the ELN criteria and the International Scale (IS) standardized definition of molecular response: MR3, BCR::ABL1 IS ≤ 0.1%; deep molecular response, MR4 or MR5, BCR::ABL1 IS ≤ 0.01% or BCR::ABL1 IS ≤ 0.001% [15].

### 2.6. Statistical Methods

The patients’ characteristics were summarized by means of cross-tabulations for categorical variables or by means of quantiles for continuous variables. Overall survival (OS; time elapsed from study entry to death) was calculated using the Kaplan–Meier product limit estimator. All analyses were performed using the SAS system software (version 9.4). The Mann–Whitney test was performed to evaluate the difference in BCR::ABL1 copies according to the positivity of S.I. (≥1.8). Study data were collected and managed using Redcap electronic data capture tools hosted at the GIMEMA Foundation [16].

## 3. Results

### 3.1. Characteristics of Patients

The main clinical characteristics of the patients are shown in Table 1. From June 2007 to October 2011, overall, 109 patients from 15 hematology centers were enrolled: 68 patients in the b3a2 vaccination trial and 41 in the b2a2 trial. The median age of the whole population at diagnosis and enrollment was 43 (17–73) and 50 (23–77) years, respectively. There were 70 males (64%) and 39 females (36%). In total, 68 (62.3%), 31 (28.4%) and 10 (9.1%) patients were stratified as low, intermediate, and high Sokal risk at presentation. Forty-two (38.5%) patients were pre-treated with interferon (IFN-α) before imatinib treatment for a median time of 23,5 months (range 1–100 months). At the time of enrollment, all patients were on imatinib for a median time of 59 months (range 18–120 months) at the dose of 400 mg/day. At study entry, 13 patients (12%) were in CCyR but not in major molecular remission (MR3), 67 (61.4%) were in MR3, 29 (26.6%) in deep molecular response (DMR); the molecular status according to BCR::ABL1 transcript is detailed in Table 1. All 109 patients completed the vaccination study core (immunization, reinforcement, and maintenance phase) and were assessable for response. Regarding further maintenance vaccine boosts, 74/109 (68%) patients received a median of one additional 6-month vaccination (range 1–3).

### 3.2. Stimulation Index Results

At baseline, before vaccination, no patients showed peptide-specific CD4+ T cell proliferation (S.I. < 1.8). After six vaccinations, 36/68 (53%) b3a2- and 19/41 (46.3%) b2a2-vaccinated patients showed a positive S.I., with an overall response of 50.5% (55/109).

At the end of the reinforcement phase (+9 vaccinations), a S.I. ≥ 1.8 was documented in 36/68 (53%) b3a2- and 22/41 (53.6%) b2a2-vaccinated patients (overall response rate 53.2%). At last evaluation, after 11 vaccinations, the overall positive immune response rate was 54%, (37/68—54.4% for b3a2 patients and 19/41—46.3% for b2a2 patients). These data demonstrate that an immune response was acquired after a minimum of six vaccinations and was then maintained along the entire study core (1 year). Of note, a positive peptide-specific CD4+ T cell proliferation index was documented at least one time in 80.8% of b3a2 and 78% of b2a2 patients. Regarding the peptide-specific immunological response in those 42 patients treated with IFN-α before imatinib, 35/42 (83.3%) and 25/42 (52.5%) achieved a measurable CD4+ T cell response at least one time and ≥2 times during the study core, respectively.

### 3.3. Molecular Responses After Vaccination

As shown in Table 2, at completion of the immunization and reinforcement schedule (6 + 3 vaccinations) 22/109 (20.2%) patients achieved and confirmed more than 1-log reduction in molecular transcript level from baseline, 14/109 (12.8%) achieved and confirmed less than a 1-log reduction, while the majority of the patients (62/109, 57%) presented a stable disease with insignificant fluctuation in molecular values. Considering the response according to molecular transcript type, 16/68 (23.5%), 9/68 (13.2%) and 38/68 (55.8%) of b3a2-vaccinated patients and 6/41 (14.6%), 5/41 (12.2%) and 24/41 (58.5%) of b2a2-vaccinated patients achieved more than 1-log reduction, less than 1-log reduction or molecular fluctuations, respectively. The molecular results were similar in patients pre-treated with INF-α (*p* > 0.05). The molecular response rate after completion of the vaccination program (immunization + reinforcement phase) of 42/109 (38.5%) IFN-α pre-treated patients and the correlation with the molecular transcript type are detailed in Table 2. Overall, only 10% of patients (11/109) had an increased molecular residual disease level of more than 1-log after vaccinations (Table 2); however, no patient switched to other TKI during the study core (12 months). When we clustered the patients according to the presence or absence of a positive peptide specific immune response (i.e., S.I. ≥ 1.8) and correlated it with the level of the BCR::ABL1 transcript achieved at the end of the study core, a statistical significant correlation between a reduction in fusion transcript and a positive peptide specific immune response was observed (*p* = 0.03) (Figure 2).

### 3.4. Vaccine Safety

A total of 746 vaccinations with CMLVAX100 and 451 vaccinations with CMLVAXb2a2-25 were administered within the 1-year study core (immunization, reinforcement, and maintenance phase). No grade 3 or 4 toxicity was observed. Grade 1 injection site skin reaction was recorded in 58/109 (53%) patients (mainly redness and mild local inflammation). Grade 2 hypertension and bronchospasm was recorded in 1/109 patients after the third CMLVAX100 vaccination. However, this patient completed the entire vaccination schedule (a total of 11 peptide vaccinations) without any further general or local adverse reaction.

### 3.5. Long-Term Follow-Up

The median follow-up is 18 years from diagnosis and 10 years from completion of study core. In total, 97/109 (89%) patients are alive, whereas 12/109 (11%) died of CML-unrelated reasons (Figure 3). Overall, 77 out of the 97 (79.4%) alive patients are still in TKI treatment (54 imatinib, 15 nilotinib, 6 dasatinib, 1 bosutinib, and 1 asciminib). Along the years, 65/109 (60%) patients discontinued imatinib: 27/109 (24.8%) for attempting TFR, 25/109 (23%) for the onset of resistance, 3/109 (2.7%) for enrollment in a clinical trial, 1/109 (0.9%) for intolerance, and 9/109 (8.2%) for CML-unrelated death, of which 3/9 (33%) were due to second neoplasia. A total of 29 patients out of 109 (26.6%) received a second TKI: in particular, 19/29 patients (65.5%) received nilotinib, 9/29 patients (31%) dasatinib and 1/29 patients (3.5%) ponatinib.

### 3.6. Treatment-Free Remission After Vaccination

In total, 34 (25 with b3a2 and 9 with b2a2 transcript) out of 109 vaccinated patients (31%) attempted to discontinue TKI treatment after achieving a stable deep molecular response (DMR). TFR was attempted after a median time of 40.5 months from the last vaccination. A total of 27/34 (79%) were always treated with imatinib (median imatinib treatment duration before discontinuation 151 months—range 50–235), while 3/34 (9%) reached DMR after switching to second-line dasatinib and 4/34 (12%) after second-line nilotinib.

Of note, 17/34 (50%) patients, who attempted TFR over time, started peptide vaccinations in MR3 or less, and 17/34 (50%) had already achieved a >MR3 at the baseline of this study. In addition, 20 out of 34 (59%) patients maintained the molecular response and are still in TFR after a median time of 48 months (range 3–190): 18/27 (66%) of those treated with imatinib and 2/4 (50%) of those switched to nilotinib.

In contrast, 13/34 (38%) patients lost the response after a median time of 7.5 months (range 3–40 months) and restarted the same TKI treatment, regaining a complete molecular response. Of note, the median value of S.I. (measuring peptide-specific immune response) checked at the end of the vaccination study core in the cohort of patients who maintain TFR was 4.5 (range 1.8–48), while the median S.I. of the whole cohort was 2.6 (1.8–48).

## 4. Discussion

With approximately 18 years of median follow-up from diagnosis of CML and 10 years of median follow-up after vaccination, the 109 patients included in these two prospective phase II studies represent the largest antitumor vaccine trial in CML and with the longest follow-up. The analysis of clinical, molecular, and immunologic data collected in this p210-derived peptide vaccine program offers an opportunity to definitely evaluate the short- and long-term impact of an active specific immune-mediated therapeutic approach in CML. At the same time, the long-lasting safety data accrued from this study furnish a solid platform to reconsider this alternative target therapy at a different timing of the CML course and with a different perspective. Our program of specific vaccination in imatinib-treated patients revealed an OS at 18 years of 89% (Figure 3), with 54/109 (49.5%) of patients still on treatment with the same drug and with 18/109 (16.5%) achieving a stable TFR. We are aware that this vaccination treatment was not a first-line treatment and thus the OS data cannot be directly compared to those of international trials showing long-term data of imatinib treatment, such as the IRIS trial [17] (OS at 10 years 83%; 49% of patients still imatinib) or the German Study IV [18] (OS at 10-year of 82% with 47% of patients still on imatinib). However, even if the patients had been on imatinib for a median of 59 months at vaccine trial entry, 73% of them had achieved only MR3 or less.

In this vaccination study, we employed two different peptide vaccines specifically developed for b3a2 and b2a2 CML patients. Indeed, the peptide-specific immune responses induced by vaccinations were comparable for b3a2 and b2a2 patients; overall, about 80% of them showed peptide-specific CD4+ T cell proliferation in vitro at least once, with no difference between the two vaccine types. Additionally, about 50% of them maintained a measurable and stable peptide-specific immune response during the entire study core, again regardless of the type of transcript and related peptide vaccine employed. These data confirmed our previous observations showing that imatinib-treated patients are capable of presenting fine T-cell immunity even against weak immunogens like tumor-derived peptides, thus contrasting with the hypothetical immunosuppressive effect of imatinib, which indeed was never shown to be clinically relevant.

Regarding the short-term impact on the immune response induced by vaccinations, it has to be noted than when we clustered the patients according to the presence or absence of a positive peptide-specific immune response (i.e., S.I. > 1.8) and correlated it with the level of BCR::ABL1 transcript achieved at the end of the study core, a statistically significant correlation between a reduction in fusion transcript and a positive peptide-specific immune response was observed. Another issue relates to the role of IFN-α, considering its known immunological and antiproliferative effect especially evident in CML. A substantial number of patients entering vaccine trials (42 patients, 38.5%) were treated with IFN-α before starting imatinib; however, according to the inclusion criteria, they still showed measurable CML before starting vaccination. No difference in terms of molecular and immune response after vaccinations was seen when compared to patients IFN-α naïve, thus suggesting, from an immunological point of view, that the anti-leukemic effect of IFN-α may have different targets than p210. The combination of imatinib and vaccination was shown to be safe: no grade 3/4 adverse events were observed and only grade 1 skin reaction at the site of injection was revealed in 53% of patients.

In recent years, we and other authors investigated the potential involvement of the immune system in the control of CML, and several immunotherapeutic strategies have been proposed [19,20]. In fact, peptide-derived antitumor vaccines have been developed with the intention of obtaining a specific antitumor response able to control and even possibly eradicate residual disease. As proof of principle, we previously reported the case of a b2a2 CP-CML patient, in CCyR and DMR after IFN-α only, who incurred an increasing molecular relapse (with rapid loss of MMR) 2 years after interferon discontinuation. At the time of relapse, the patient was treated only with a b2a2 breakpoint-derived peptide vaccine. After vaccinations, she developed an adequate peptide-specific CD4+ T-cell response and she obtained a stable MR5 still lasting today after 17 years without any further CML therapy [21]. Other authors have tested breakpoint-derived peptide vaccines in CML, attempting to increase immunogenicity by using synthetic structures (“heteroclitic” peptides) [22,23] or by linking P210 breakpoint peptides to known immunogenic PAN DR epitope [24]. Although only a small series of patients were included in these trials, most patients developed T-cell responses to BCR::ABL1 peptides and some patients improved the molecular response after vaccination with no serious side effects observed [22,23,24]. To our knowledge, no long-term follow-up of CML patients included in all the above or in other BCR::ABL1 peptide vaccine trials is reported in the literature. Since the first evidence by Mahon et al. that imatinib-treated patients who reached a sustained DMR may discontinue TKI treatment without loss of molecular response, TFR became an important endpoint for CML patients [25,26]. Indeed, several reports, mainly including imatinib-treated patients, confirmed in about 40–60% of patients the achievement of a stable TFR if discontinuation occurs after a sustained deep molecular response and a long-lasting TKI treatment [27,28]. In our study, when considering vaccinated patients treated only with imatinib, 27/109 (25%) attempted discontinuation and 18/27 (66%) remained disease-free with a medium follow-up of 48 months. Despite undergoing long-lasting imatinib treatment before TKI discontinuation (median 171 months), 44% of them (12/27) started vaccinations after achieving MR3 or less, and, thus, the observed rates of TFR (closer to those observed after second-generation TKI treatment) may have been increased by the immune effect of CML peptide vaccination as the median S.I. was higher in patients who maintained TFR than in the whole cohort (4.5 vs. 2.6).

Overall, the results obtained with this breakpoint peptides vaccination program and the high safety of this immunotherapeutic tool are in favor of reconsidering a similar approach in CML patients. Indeed, the evidence that the induction of a P210-specific immune response correlates with the reduction in BCR::ABL1 transcript may “revitalize” a vaccination strategy. Although no clear influence of p210 breakpoint peptide vaccines emerges in terms of the success rates of TFR from the analysis of the long-term follow-up, the excellent data on tolerability, safety, and positive immunologic stimulation may justify further studies on new vaccines with the aim of improving the effectiveness of current therapies. Alternatively, this strategy could be proposed after the discontinuation of TKIs to stimulate the immunological compartment against the residual disease again with the aim to increase the rate of success after stopping treatment. The impressive results of COVID-19 mRNA vaccines fighting the recent coronavirus pandemic have led to significant enthusiasm and an important burst for mRNA vaccines in the pharmaceutical and biotechnology industries [29]. Thus, several clinical trials are underway to prove the efficacy and safety of mRNA vaccines in patients with various types of cancer [30].

## 5. Conclusions

In the current scenario, we believe that pivotal b2a2 and b3a2 peptide therapeutic vaccine trials, albeit with their limitations (such as a lack of a randomized control), may be reconsidered and reevaluated. The feasibility of such an active immunotherapeutic approach with the absence of off-target events, high OS and a not-negligible rate of successful TFR could prompt newer vaccine strategies, such as mRNA BCR::ABL1 breakpoint vaccines for low-disease-burden CML patients in order to improve the current therapeutic path towards a cure.

## Figures and Tables

**Figure 1 vaccines-13-00419-f001:**
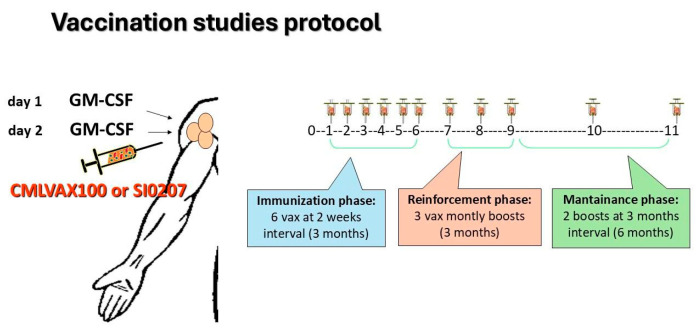
CMLVAX100 and SI0207 peptide vaccine protocols. Peptide vaccination schedule consisted of 6 vaccinations at 2-week intervals (“immunization” phase) followed by 3 monthly boosts of the vaccine (“reinforcement” phase) and, subsequently, by 2 additional boosts at 3-month intervals (“maintenance” phase).

**Figure 2 vaccines-13-00419-f002:**
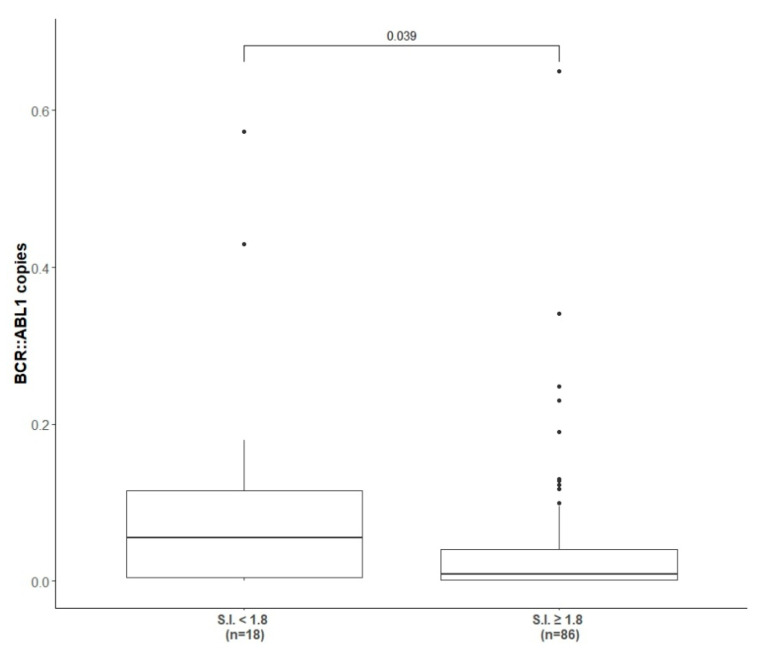
Correlation between BCR::ABL1 copies and presence of positive peptide immune response. The correlation between the level of BCR::ABL1 transcript at the end of study core (1 year) and the presence (at least one S.I. ≥ 1.8) or absence of peptide specific immune response induced by peptide-derived vaccinations.

**Figure 3 vaccines-13-00419-f003:**
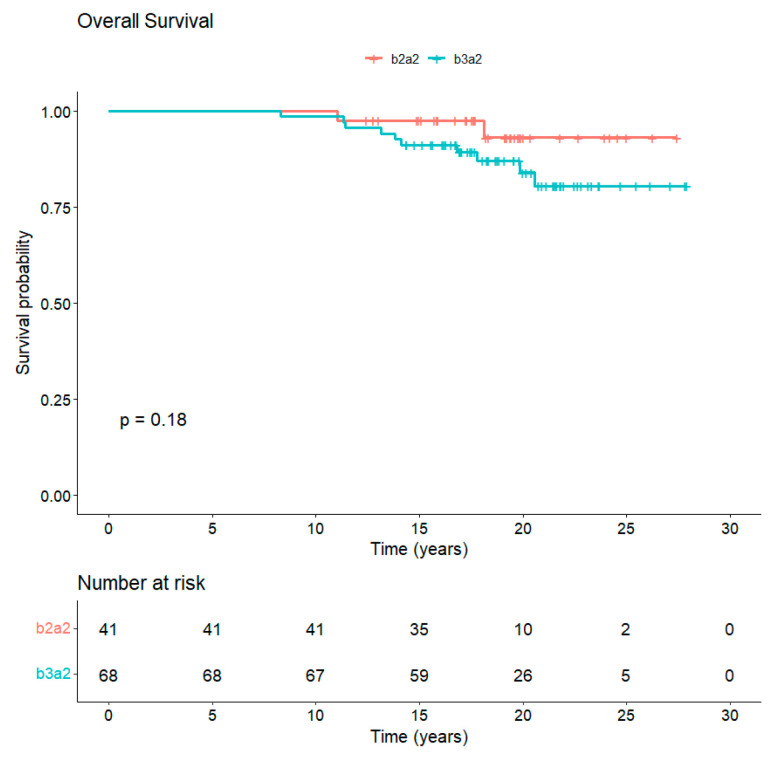
Overall survival of CML patients vaccinated with b3a2- and b2a2-derived peptide vaccines. The median follow-up of CML patients at 18 years from diagnosis is 89% with no statistical differences between b3a2 and b2a2 vaccination.

**Table 1 vaccines-13-00419-t001:** Patients’ characteristics at enrolment into vaccine trials.

		Total	b3a2	b2a2
Patients		109	68	41
Median age (yr) at the diagnosis (range)		43 (17–23 yr)	43 (22–73 yr)	50 (17–69 yr)
Median age (yr) at the enrollment (range)		50 (23–77 yr)	51 (28–77 yr)	46 (23–77 yr)
Sex	male	70 (64%)	44 (64.7%)	26 (63.4%)
	female	39 (36%)	24 (35.3%)	15 (36.6%)
Sokal	high	10 (9.2%)	4 (5.9%)	6 (14.6%)
	intermediate	31 (28.5%)	21 (30.9%)	10 (24.4%)
	low	68 (62.3%)	43 (63.2%)	25 (61%)
Disease status pre-vaccine	no MR3	13 (12%)	10 (15%)	3 (7%)
	MR3	67 (61.4%)	47 (69%)	20 (49%)
	>MR3	29 (26.6%)	11 (16%)	18 (44%)

**Table 2 vaccines-13-00419-t002:** Molecular response to peptide vaccine at completion of immunization and reinforcement schedule (6 + 3 vaccinations) according to type of BCR::ABL1 transcript and according to pre-imatinib treatment.

		Reduction >1 log BCR::ABL1	Reduction <1 log BCR::ABL1	Stable Disease with BCR::ABL1 fluctuation	Increase ≥1 log BCR::ABL1	*p* Value
Whole cohort	109	22	14	62	11	
B3a2	68	16 (23.5%)	9 (13.2%)	38 (56%)	5 (7.3%)	0.494
B2a2	41	6 (14.6%)	5 (12.2%)	24 (58.5%)	6 (14.6%)	
INF-α yes	42 (38.5%)	9 (21.4%)	7 (16.7%)	22 (52.4%)	4 (9.5%)	0.655
INF-α no	67 (61.5%)	11 (16.5%)	7 (10.4%)	42 (62.7%)	7 (10.4%)	

## Data Availability

The original contributions presented in this study are included in the article. Further inquiries can be directed to the corresponding author.

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
