# Peer review of "Therapeutic Vaccinations with p210 Peptides in Imatinib-Treated Chronic Myeloid Leukemia Patients: 10 Years Follow-Up of GIMEMA CML0206 and SI0207 Studies"

_vaccines, 2025, doi:10.3390/vaccines13040419_

Round 1
Reviewer 1 Report
Comments and Suggestions for Authors
The report by Sicuranza et al. provides long term data on a large cohort of patients who received a BCR::ABL fusion peptide vaccine along with TKI. The report is clear, provides good quality data and is useful to plan future therapeutic vaccination studies. A few clarifications and additions would nonetheless increase the quality of the manuscript:
1- Line 90. Please provide the rationale for the HLA allele chosen.
2- Line 204. Please justify the cut-off of 1.8 to define a “positive” stimulation index (SI) and explain why data on the CD8 SI were not obtained despite the presence of several MHC-class I peptides in vaccines. If available/done, the data on CD8 should be included.
3- Line 218 and following. The relationship between the presence of a demonstrable anti-peptide response and the transcript level at the end of the vaccination course (study core) is interesting but more telling would be to show the log reduction in function of peptide responses. As currently written, we don’t know whether a positive stimulation index correlates with a response (i.e. a reduction of the transcript levels). Please provide data that would indicate (or not) that a demonstrable peptide response correlates with response, stable disease or increased transcript levels.
- Discussion. Please discuss the following limitation of the study: the presentation of the peptides at the cell surface of BCR::ABL bearing cells was not demonstrated, nor the lysis of primary leukemic cells by peptide reactive cells (or cytokine secretion for CD4 cells). It remains unclear whether the leukemia cells actually express the predicted peptides. Also, alteration in peptide presentation may be a mechanism of resistance. Future studies should factor in these notions.
Author Response
Reviewer 1
The report by Sicuranza et al. provides long term data on a large cohort of patients who received a BCR::ABL fusion peptide vaccine along with TKI. The report is clear, provides good quality data and is useful to plan future therapeutic vaccination studies. A few clarifications and additions would nonetheless increase the quality of the manuscript:
- Line 90. Please provide the rationale for the HLA allele chosen.
As reviewer suggested, we added in the introduction a brief rationale for the HLA allele chosen as follows: “Indeed, after screening the ability of a series of synthetic peptides corresponding to the junction sequences of p210 protein, to bind according to putative anchor motifs class I and class II molecules (i.e. HLA A1, A2.1, A3.2, B8 and DR11) we and others identified some b3a2 and b2a2 breakpoint peptides able to elicit in vitro a specific T cell response both in normal donors and in CML patients [9,10].”
Further details may be found in both references:
- Bocchia, M.; Wentworth, P.A.; Southwood, S. et al. Specific binding of leukemia oncogene fusion protein peptides to HLA class I molecules. Blood. 1995; 85: 2680-2684.
- Bocchia, M.; Korontsvit, T.; Xu, Q. et al. Specific human cellular immunity to bcr-abl oncogene derived peptides. Blood. 1996; 87: 3587-3592.
However, to reduce the percentage of self-citations, as proposed by the Editor we removed from the manuscript the ref. Bocchia M, Blood 1995 as in Bocchia et al Blood 1996 the rationale of the choice of peptides is summarized (it would be redundant to add also this ref.)
2- Line 204. Please justify the cut-off of 1.8 to define a “positive” stimulation index (SI) and explain why data on the CD8 SI were not obtained despite the presence of several MHC-class I peptides in vaccines. If available/done, the data on CD8 should be included.
Our previous in vitro studies demonstrated that the CML peptides were able to induce specific cytotoxic T lymphocytes (CD8) and specific T cell proliferation (CD4) (Bocchia et al. Blood 1996). However, given the difficulties to check for direct CD8 T cell vs CML cytotoxicity in the context of low tumor burden/minimal residual disease as the trial was designed, it was impossible. As such, we decided to measure peptide specific T cell activation following the vaccine by performing an easier, more reproducible CD4 T cell specific proliferation assay. Previous experiments (Bocchia et. al Blood 1996, Bocchia et. al Lancet 2005) demonstrated a positive result when freshly magnetic beads isolated CD4 T cells increase at least by 1.8 times after the incubation with class II peptides included in the vaccine. Indeed, before vaccination, no proliferation of CD4+T cells after incubation with b3a2 or b2a2 peptides was ever documented.
3- Line 218 and following. The relationship between the presence of a demonstrable anti-peptide response and the transcript level at the end of the vaccination course (study core) is interesting but more telling would be to show the log reduction in function of peptide responses. As currently written, we don’t know whether a positive stimulation index correlates with a response (i.e. a reduction of the transcript levels). Please provide data that would indicate (or not) that a demonstrable peptide response correlates with response, stable disease or increased transcript levels.
The main reason why we didn’t show per each patient the correlation between S.I. and molecular response, it was because this would overload the manuscript with redundant data (considering over hundreds of patients and at least 4 SI evaluations per patient). However, we think that the summary of the molecular response (in terms of log reduction) at the end of the vaccine protocol in Table 2 and the correlation between molecular response and positive stimulation index represented in figure 2, may explain the outcome after vaccinations as well answer the reviewer's query.
4- Discussion. Please discuss the following limitation of the study: the presentation of the peptides at the cell surface of BCR::ABL bearing cells was not demonstrated, nor the lysis of primary leukemic cells by peptide reactive cells (or cytokine secretion for CD4 cells). It remains unclear whether the leukemia cells actually express the predicted peptides. Also, alteration in peptide presentation may be a mechanism of resistance. Future studies should factor in these notions.
We thank the reviewer for arising all these issues that were indeed at least partly answered in previous publications. Regarding the demonstration of peptides on the MHC I class molecules onto CML cells, this was elegantly demonstrated in the publication of Clark RE et al. (Clark RE, Dodi IA, Hill SC, et al. Direct evidence that leukemic cells present HLA-associated immunogenic peptides derived from the BCR-ABL b3a2 fusion protein. Blood. 2001 Nov 15;98(10):2887-93. doi: 10.1182/blood.v98.10.2887.) We added this useful information in the introduction (line 93).
Regarding the lysis of primary CML leukemia cells by peptide primed T cells this was not possible to demonstrate for 2 main reasons:
1) Given the characteristics of CML, which consists of blood circulating cells at different stages of maturation, but mainly neutrophils and progenitors, it is very difficult to sort a homogenous population to be used as target of T cell mediated lysis. Moreover, we think that the main target of a peptide-induced T cell response should be the CML leukemic stem cells, which are very rare even at diagnosis and thus very difficult to isolate and use as targets. Only in blastic crisis it would be possible to purify enough cells to set up direct cytotoxicity assays, but none of our patients progressed to a blastic crisis.
2) Considering the above, the possibility to set up a cytotoxicity assay in the patients included in the trial was impossible as the burden of disease at study entry was very limited.
Moreover, given the aim of this clinical study and considering all previous publications on the topic we are not confident that going into experimental details as those raised by the reviewer with his observations would be of any additional value for the readers. We thought that the reported case of a CML patient losing her response after 2 years from IFN discontinuation and achieving a complete molecular response, still ongoing, after peptide vaccination only (with a strong peptide-specific CD4+ T cell response) is a good proof of principle may be even beyond in vitro data (ref. Bocchia et al. Nat Rev Clin Oncol 2010). In addition, we thought that for a therapeutic vaccine it is may be important to show immunogenicity and specific T cell priming and subsequently follow disease outcome (a bit similarly with infectious disease vaccines, in which the goal is inducing antibodies response and monitoring afterwards the rate of disease into the population to demonstrate prevention efficacy).
Reviewer 2 Report
Comments and Suggestions for Authors
Thank you for the opportunity to review your valuable paper.
This is a 10-year long-term follow-up report on the safety and efficacy of the p210 derived peptide vaccine for patients with CML.
With the advent of TKIs, survival outcomes in patients with CML has dramatically improved and CML is the most successful form of the disease treated by TKIs, ushering in today's trends in molecular targeted therapy medicine. Although survival outcomes have improved dramatically, the most important question now is whether patients with CML can maintain a complete cure after discontinuation of TKIs. In this regard, the study of the efficacy of peptide vaccines as immunotherapy described in this paper is valuable and provides an important issue to compensate for TKIs therapy
I have a few comments and questions regarding the content and will present them.
Although it is acceptable to continue TKI treatment during peptide vaccination in this study, it is noted in the discussion that 27 patients attempted to discontinue TKI treatment during peptide vaccination and 18 of them maintained TFR in the present study. The information on the TKI discontinuation group is very important and should be presented in detail in the results rather than mentioned in the discussion.
The cell proliferation response of CD4-positive cells in vitro has been investigated as one way to determine the efficacy of peptide vaccines. In examining the in vitro immune response to a specific immune antigen, the cell proliferation response using 3H-thymidine uptake does not appear to be a sensitive assay, and it is questionable whether it accurately reflects the immune response. When considering a study such as this one, the IFN-γ release assay usually should be considered. It would be better to discuss this point or have data to provide.
Author Response
Reviewer 2
Thank you for the opportunity to review your valuable paper.
This is a 10-year long-term follow-up report on the safety and efficacy of the p210 derived peptide vaccine for patients with CML.
With the advent of TKIs, survival outcomes in patients with CML has dramatically improved and CML is the most successful form of the disease treated by TKIs, ushering in today's trends in molecular targeted therapy medicine. Although survival outcomes have improved dramatically, the most important question now is whether patients with CML can maintain a complete cure after discontinuation of TKIs. In this regard, the study of the efficacy of peptide vaccines as immunotherapy described in this paper is valuable and provides an important issue to compensate for TKIs therapy
I have a few comments and questions regarding the content and will present them.
Although it is acceptable to continue TKI treatment during peptide vaccination in this study, it is noted in the discussion that 27 patients attempted to discontinue TKI treatment during peptide vaccination and 18 of them maintained TFR in the present study. The information on the TKI discontinuation group is very important and should be presented in detail in the results rather than mentioned in the discussion.
In order to be clearer, as suggested by the reviewer, we rephrased the paragraph (3.6) regarding CML patients who discontinued TKI. However, it has to be noted that patients discontinued TKI well after the end of vaccination protocol (after a median time of 40 months from last vaccination) as stated in the paragraph.
The cell proliferation response of CD4-positive cells in vitro has been investigated as one way to determine the efficacy of peptide vaccines. In examining the in vitro immune response to a specific immune antigen, the cell proliferation response using 3H-thymidine uptake does not appear to be a sensitive assay, and it is questionable whether it accurately reflects the immune response. When considering a study such as this one, the IFN-γ release assay usually should be considered. It would be better to discuss this point or have data to provide.
Thank you for this interesting observation. Although the IFN-γ release assay is a valid approach useful to evaluate the immune response of T lymphocytes and it was also employed in our previous pivotal experiments, for this trial we decided to continue to perform the evaluation of peptide-specific immune response with standard 3H-thymidine incorporation assay to reproduce standardized data comparable to those described in our pivotal trial published on Lancet in 2005.
We agree with the reviewer that nowadays 3HT incorporation assay is not the perfect assay to measure T cell proliferation. However, it has to be considered that all in vitro data were performed in the years 2005-2010 and the 3HT assay was chosen in order to be consistent (and comparable) to assays performed even earlier as 2000-2005.
Given these premises, we hope the reviewer would agree that including a sentence stating why we did not use IFN-y release assay it would not add valuable information for the readers in the context of the present study.
Reviewer 3 Report
Comments and Suggestions for Authors
Because quiescent chronic myeloid leukemia (CML) stem cells survive despite inhibition of chimeric BCR::ABL1 fusion protein with tyrosine kinase inhibitors (e.g., imatinib) in CML patients, an alternative immune-based targeted therapy has been hypothesized and developed. In this article, the 10-year follow-up results of two multicenter peptide vaccine phase II studies, GIMEMA CML0206 and SI0207, respectively against the b3a2 and b2a2 sequence of p210, the oncogenic protein derived from the BCR::ABL1 fusion gene in 109 imatinib-treated CML patients are presented. Overall, the excellent data on feasibility, safety, absence of off-target events, high overall survival (OS) and substantial rate of treatment-free remission may justify further studies on new vaccines (e.g., mRNA vaccines) with the aim of improving the effectiveness of current therapies.
This manuscript presents a well-written and informative overview of the clinical, molecular, immunologic and survival data of two phase II p210-derived peptide vaccine programs for chronic phase CML patients treated with imatinib and with persistence of molecular residual disease. While the focus of the study is on evaluating the impact of the vaccine trials on selected endpoints, the study design apparently did not include direct comparison with CML patients receiving long-term imatinib treatment alone. Could comparison be made with historical data from long-term imatinib-treated CML patients at the same medical institutions? The authors acknowledge that OS data cannot be directly compared to that of international trials (line 306-310).
Line 53. Is short-term after the first 3 months of vaccination? Please be specific.
Line 91. What is QS-21?
Line 109-111. How many centers were involved? Were the laboratory analyses standardized across all centers or were they centralized at one or a few centers? What were the inter-laboratory assay variability?
Line 146-160. At which time points after vaccinations were the immune response analyzed? How were the CD4+ T cells purified? Explain the principle of the assay.
Line 171-174. Rewrite as “BCR::ABL1IS” as “BCR::ABL1 IS” to make this acronym readable.
Line 215, 227. What were the dose and duration of interferon (IFN)-α treatment?
Line 248-250. Were any patients lost to follow-up during the study period? What were the “CML-unrelated reasons”?
Line 269-272. After how many months did patients lose the response? What were their stimulation index values (SI) after regaining a complete molecular response?
Fig. 2. Please report the correlation results for the b3a2 and b2a2 CML patients separately. Do the boxplot lines correspond to the median, 25th and 75th percentiles, and what else? Indicate the number of patients per group and p-value(s).
Fig. 3. The axis labels and survival data lines are difficult to read.
Supplementary Material and Methods. Second paragraph: Does the vaccine consist of 100 µg of one single peptide? Third paragraph: Was the CMLVAX100 also approved by the Istituto Superiore di Sanita?
Author Response
Reviewer 3
Because quiescent chronic myeloid leukemia (CML) stem cells survive despite inhibition of chimeric BCR::ABL1 fusion protein with tyrosine kinase inhibitors (e.g., imatinib) in CML patients, an alternative immune-based targeted therapy has been hypothesized and developed. In this article, the 10-year follow-up results of two multicenter peptide vaccine phase II studies, GIMEMA CML0206 and SI0207, respectively against the b3a2 and b2a2 sequence of p210, the oncogenic protein derived from the BCR::ABL1 fusion gene in 109 imatinib-treated CML patients are presented. Overall, the excellent data on feasibility, safety, absence of off-target events, high overall survival (OS) and substantial rate of treatment-free remission may justify further studies on new vaccines (e.g., mRNA vaccines) with the aim of improving the effectiveness of current therapies.
This manuscript presents a well-written and informative overview of the clinical, molecular, immunologic and survival data of two phase II p210-derived peptide vaccine programs for chronic phase CML patients treated with imatinib and with persistence of molecular residual disease. While the focus of the study is on evaluating the impact of the vaccine trials on selected endpoints, the study design apparently did not include direct comparison with CML patients receiving long-term imatinib treatment alone. Could comparison be made with historical data from long-term imatinib-treated CML patients at the same medical institutions? The authors acknowledge that OS data cannot be directly compared to that of international trials (line 306-310).
Line 53. Is short-term after the first 3 months of vaccination? Please be specific.
Thank you for the question. The short-term results are referred to the evaluation at the end of the vaccination protocol meaning at 12 months from starting. We specified it in the abstract.
Line 91. What is QS-21?
QS-21 is the immunological adjuvant derived from the active fraction of the tree Quillaja Saponaria. We specified it in the text (line 97).
Line 109-111. How many centers were involved? Were the laboratory analyses standardized across all centers or were they centralized at one or a few centers? What were the inter-laboratory assay variability?
Thank you for the question. The two multicenter studies involved a total of 15 Hematology centers (14 Italian and 1 German); we added the number of centers in the manuscript (line 201).
Immune response assays were centralized to the Siena Hematology Lab, while molecular evaluation for the BCR::ABL transcript have been performed by each center.
When considering the same timeframe period, inter laboratory variabilities are very rare, because each laboratory belonging to the study followed the guidelines of that period. However, over the years the guidelines changed and thus BCR::ABL values have been expressed differently. Finally, since 2008 it was activated the LabNet CML Network, which promotes standardized procedures for BCR::ABL monitoring. All laboratories of hematology centers participating in the study belong to LabNet. Regarding the only German center, we centralized in Siena also the molecular assay.
Line 146-160. At which time points after vaccinations were the immune response analyzed? How were the CD4+ T cells purified? Explain the principle of the assay.
The immune responses after vaccinations were performed as follows: at the end of the immunization phase 2 weeks after the 6 vaccinations (12th week); at the end of the reinforcement phase 3 weeks after the 9th vaccination; at the end of the maintenance phase 2 weeks after the 11th vaccination.
CD4+T cell purification has been performed by using a magnetic separation with microbeads (Miltenyi Biotec). Briefly, after a separation of peripheral blood mononuclear cells (PBMCs) cells with a density gradient solution, the CD4+ cells have been magnetically labeled with CD4 Microbeads. Then, the cell suspension has been loaded on a column which is placed in the magnetic field of a separator. The magnetically labeled CD4+ cells have been retained within the column and after removing the column from the magnetic field, the magnetically retained CD4+ cells have been eluted as the positively selected cell fraction.
We added the complete description of CD4+ cells purification in the supplemental methods and also, we briefly specified it in the paragraph 2.4.
Line 171-174. Rewrite as “BCR::ABL1IS” as “BCR::ABL1 IS” to make this acronym readable.
Thank you for the observation. We rewrote BCR::ABL1 IS.
Line 215, 227. What were the dose and duration of interferon (IFN)-α treatment?
The median duration of interferon (IFN)-α treatment was 23,5 months (range 1- 100 months). We added it to the manuscript (line 207). Regarding the dose, unfortunately the exact information per each patient is not available. However, considering that all these CML patients were treated in prestigious Italian hematology centers, most of them academic, we are confident that the IFN dosage was recommended according to standard CML treatment guidelines of the time.
Line 248-250. Were any patients lost to follow-up during the study period? What were the “CML-unrelated reasons”?
No patients were lost to follow-up. The “CML-unrelated reasons" for which patients died were the followings: 1 multiple myeloma, 1 pancreatic adenocarcinoma, 1 prostate cancer, 1 chronic renal failure, 2 heart failure, 1 acute myocardial infarction, 1 senile dementia and 3 for natural senescence. For a patient the cause of death is unknown.
Line 269-272. After how many months did patients lose the response? What were their stimulation index values (SI) after regaining a complete molecular response?
The 13 patients lost the response after a median of 7,5 months from the time of TKI discontinuation. We added this information to the manuscript (line 283-284). The S.I. value after regaining a complete molecular response is not available because, according to the study design, the S.I. evaluation was only measured before, during and at the end of vaccine study core. Patients attempted to stop TKI at a median time of 40.5 months from last vaccination.
Fig. 2. Please report the correlation results for the b3a2 and b2a2 CML patients separately. Do the boxplot lines correspond to the median, 25th and 75th percentiles, and what else? Indicate the number of patients per group and p-value(s).
We tried to realize a new figure as suggested. However, we think that the graph is not very representative because the total CML patients with an S.I. <1.8 are only 18 of which only 5 belonging to the b2a2 cohort. Thus, we preferred to maintain the previous figure with cumulative data. We added p value and the number of patients per group. We confirmed that the boxplot lines correspond to the median, 25th and 75th percentiles.
Fig. 3. The axis labels and survival data lines are difficult to read.
Thank you for your observation. We realized a new clearer figure which substitutes the previous in the manuscript.
Supplementary Material and Methods. Second paragraph: Does the vaccine consist of 100 µg of one single peptide? Third paragraph: Was the CMLVAX100 also approved by the Istituto Superiore di Sanita?
As described in Supplementary method, the vaccine CMLVAX100 consists of 100 µg per each peptide included (5 peptides).
Yes, the CMLVAX100 was approved by the Istituto Superiore di Sanita and we spelled out the acronyms ISS in the supplementary method.
Reviewer 4 Report
Comments and Suggestions for Authors
This important article summarizes a 10-year follow-up vaccine study using p210 peptides in imatinib-treated chronic myeloid leukemia patients. Although there is no significant difference in overall survival, this study provides fundamental guidance for future trials. This Reviewer has identified a few points that need to be addressed:
Materials and Methods:
2.5. Molecular response evaluation: The authors shall provide more details on the RT-PCR experiments, including sample preparation, RNA isolation, PCR equipment, etc.
Results:
3.4. Vaccine safety and 3.6. Treatment-free remission after vaccination:
If possible, table(s) summarizing the data is suggested to be included.
Author Response
Reviewer 4
This important article summarizes a 10-year follow-up vaccine study using p210 peptides in imatinib-treated chronic myeloid leukemia patients. Although there is no significant difference in overall survival, this study provides fundamental guidance for future trials. This Reviewer has identified a few points that need to be addressed:
Materials and Methods:
2.5. Molecular response evaluation: The authors shall provide more details on the RT-PCR experiments, including sample preparation, RNA isolation, PCR equipment, etc.
Molecular evaluation of BCR::ABL transcripts have been performed according to the standardized procedures divulgated over the years by the experts’ panels of CML (references 13-15) and according to the Italian LabNet CML Network guidelines since its origin in 2008. Briefly, peripheral blood samples from CML patients have been centrifuged to obtain the buffy coat from which the mRNA has been extracted. Then, reverse transcriptase quantitative PCR (RT-qPCR) has been performed to estimate the amount of BCR::ABL1 relative to an internal reference gene, most commonly ABL1, or GUSB. The results were expressed on an International Scale (IS) as a percentage. Expression of results on the IS depends on each testing laboratory either having obtained a laboratory-specific conversion factor (CF) by sample exchange with an established reference laboratory or by using kits and reagents that have been calibrated to the World Health Organization International Genetic Reference Panel for quantitation of BCR::ABL1 mRNA.
Given that the evaluation of molecular response in CML is a well-established assay as reported above, we preferred to go into deeper details only in the supplemental methods (see putative file).
Results:
3.4. Vaccine safety and 3.6. Treatment-free remission after vaccination:
If possible, table(s) summarizing the data is suggested to be included.
We thank the reviewer for this suggestion. We think that the data regarding TFR attempt and follow-up are sufficiently detailed and described along the specific paragraph (3.6) and we are afraid that an additional table would not be of easy interpretation from the readers. However, to comply with the suggestion of the reviewer we attempted to design a flow-chart (see below) and we ask the Editor and the reviewer himself if this would be of help to better clarify this part of the paper. Regarding vaccine related toxicity given the limited AEs (mainly grade 1 local skin redness) we propose an explanatory graph (see below) and not an additional table. Even in this case we kindly ask for the opinion of the Editor and the reviewer.
Round 2
Reviewer 2 Report
Comments and Suggestions for Authors
I would like to express my gratitude for the opportunity to review the revised version.
Thank you for your sincere response to my previous questions.
I think that the content and purpose of the paper is generally acceptable.